# Integrated Expression Analysis of Small RNA, Degradome and Microarray Reveals Complex Regulatory Action of miRNA during Prolonged Shade in Swarnaprabha Rice

**DOI:** 10.3390/biology11050798

**Published:** 2022-05-23

**Authors:** Madhusmita Panigrahy, Kishore Chandra Sekhar Panigrahi, Yugandhar Poli, Aman Ranga, Neelofar Majeed

**Affiliations:** 1Biofuel & Bioprocessing Research Centre, Institute of Technical Education and Research, Siksha ‘O’ Anusandhan University, Bhubaneswar 751002, India; 2National Institute of Science Education and Research, Homi Bhabha National Institute (HBNI), Khurda 752050, India; aman.ranga@niser.ac.in (A.R.); nelofarmajeed72@gmail.com (N.M.); 3ICAR-Indian Institute of Rice Research, Rajendra Nagar, Hyderabad 500030, India; yugandhar12345@gmail.com

**Keywords:** Swarnaprabha, rice, shade, low-light, microRNA, yield, shade avoidance, shade tolerance

## Abstract

**Simple Summary:**

Significant progress has been made to understand shade and low-light due to their deteriorative impact on plant growth and crop yield. The authors of the present study identified novel microRNA (miR) regulations for tolerant phenotypic responses associated with a sustainable yield of Swarnaprabha, a shade-tolerant rice genotype. We present three new findings, two of which are demonstrations of novel concepts. First, we show that the differentially expressed miRs had drastic differences in the category of targets based on their function and pathway. Second, we show that neutrally regulated and uniquely expressed miRs can also contribute to the shade-tolerance response by altering the differential expression of their targets under different light conditions. Third, we identified the interactions of 16 miRNA and 21 target pairs, whose actions can significantly contribute to the shade-tolerance phenotype and sustainable yield of Swarnaprabha rice. The findings of this study will significantly update the knowledge of the epigenetic regulations of shade tolerance in rice. The authors of further studies can use the genomic data and findings of this study to generate shade-tolerant crop varieties to combat prevailing unpredictable weather conditions in order to maintain food security.

**Abstract:**

Prolonged shade during the reproductive stage can result in significant yield losses in rice. For this study, we elucidated the role of microRNAs in prolonged-shade tolerance (~20 days of shade) in a shade-tolerant rice variety, Swarnaprabha (SP), in its reproductive stage using small RNA and degradome sequencing with expression analysis using microarray and qRT-PCR. This study demonstrates that miRNA (miR) regulation for shade-tolerance predominately comprises the deactivation of the miR itself, leading to the upregulation of their targets. Up- and downregulated differentially expressed miRs (DEms) presented drastic differences in the category of targets based on the function and pathway in which they are involved. Moreover, neutrally regulated and uniquely expressed miRs also contributed to the shade-tolerance response by altering the differential expression of their targets, probably due to their differential binding affinities. The upregulated DEms mostly targeted the cell wall, membrane, cytoskeleton, and cellulose synthesis-related transcripts, and the downregulated DEms targeted the transcripts of photosynthesis, carbon and sugar metabolism, energy metabolism, and amino acid and protein metabolism. We identified 16 miRNAs with 21 target pairs, whose actions may significantly contribute to the shade-tolerance phenotype and sustainable yield of SP. The most notable among these were found to be *miR5493*-*OsSLAC* and *miR5144*-*OsLOG1* for enhanced panicle size, *miR5493*-*OsBRITTLE1-1* for grain formation, miR6245-*OsCsIF9* for decreased stem mechanical strength, *miR5487*-*OsGns9* and *miR168b*-*OsCP1* for better pollen development, and *miR172b*-*OsbHLH153* for hyponasty under shade.

## 1. Introduction

Rice cultivation is subjected to significant reductions in light irradiation (>500 µm m^−1^ s^−1^) due to overcast clouds and rain during kharif season in many countries in Southeast Asia. Decreased light irradiance can affect all stages of rice plant growth, but the severity is highest (accounting for nearly 40–50% of yield loss) when low light is imposed during the reproductive stage. This is caused by reduced photosynthesis vis-a-vis dry weight [1] associated with decreases in panicle and spikelet numbers. Though genotypic variance accounts for up to ~39% of total yield variation, heritability can reach up to 58% under shade [2]. Indica rice varieties such as Swarnaprabha (SP), Purnendu, and Sashi exhibit efficient shade-tolerant mechanisms with sustainable yield under prolonged shade [2,3]. These genotypes can attain sustainable yields under shade by maintaining higher panicle numbers [4], higher grain numbers [5], higher net photosynthesis rates (P_n_) [6], better translocation efficiencies from source to sink organs, nutrient remobilization for growing organs via autophagy, and the recycling of chloroplast and root plastids [7]. The plethora of shade-tolerant mechanisms that allow for sustainable yields was also shown to include higher rates of panicle emergence and antioxidant activities [8], thereby restricting membrane damage.

MicroRNAs (miRs) are noncoding RNAs of 21–24 nucleotides in length, known in different plant species for their significant roles in various developmental processes [9] including floral transition [10], the control of flowering time [11,12], and floral organ development [13,14]. In recent decades, miRNA-mediated post-transcriptional gene silencing (PTGS) via cleavage or translational repression has been shown to mediate several photomorphogenic events including phytochrome B (PhyB)-mediated light signaling [15], seedling de-etiolation [16], cotyledon opening, and photoperiod-dependent flowering [17]. miRNAs modulate responses to various environmental stresses in plants [18,19,20]. Responses to abiotic stresses such as drought salinity, cold, and phosphate deficiency involve the regulatory actions of *miR156*, *miR160*, *miR397*, *miR319*, *miR402* and *miR399*, as has been demonstrated in different plant species including rice [21,22,23,24,25,26,27]. Recent CRISPR-Cas9 approaches to generate mutants of miRNA or their targets in rice have given hints that miRNA can serve as a crucial tool in generating several important agricultural traits for crop improvement [28,29]. The miR regulation for high yields in recombinant inbred lines (BILs) of rice has demonstrated the involvement of ethylene and auxin signaling pathways. *miR2877*, *miR530-5p*, and *miR396h* were found to positively affect grain filling in the apical spikelets of the high-yield recombinant inbred lines of rice by controlling the expression of their targets [30]. *miR156* and *miR172* play consistent roles in developmental transitions in both monocots and dicots [31]. Most studies have shown that miRNA-mediated actions revolve around transcriptional events and transcription factors [32]. PhyB-mediated signaling in a *phyB* mutant background was shown to focus on seven different types transcription factors (TFs) that were targeted by miR [15]. The *PHYTOCHROME INTERACTING FACTOR* (*PIF*) TFs have also been shown to suppress the expression of *miR156* [33]. miR regulation has also been reported to vary from species to species, such as the case of *miR172*. Specifically, *miR172* expression was observed in a *phyB* mutant in rice but not detected in potatoes [34], indicating the need for the verification of miR-mediated signaling in each species.

The degradation of the *HYPONASTIC LEAVES* (*HYL1*) miRNA biogenesis factor leads to the onset of gene silencing during shade, and the dephosphorylation of HYL1 leads to the reactivation of miRNA biogenesis to maximize light uptake [35]. The *miR156*-mediated downregulation of *SQUMOSA PROMOTOR BINDING PROTEIN-LIKE* (*SPL*) genes is required for end-of-day far-red (EOD-FR) responses and an increase in PIFs for a portion of low (red) R:FR responses [33]. Under prolonged shade, SP plants have shown several tolerant mature plant phenotypes including higher rates of panicle emergence, flag leaf lengths, panicle lengths, grain weight/panicle, and percent grain filling. Decreases in plant yield under prolonged shade were found to be less in SP compared to other genotypes under study. A genome-wide expression analysis showed the upregulation of most of the ethylene and cytokinin pathway genes including *ETHYLENE RESPONSIVE BINDING PROTEIN 2* (*EREBP2*), *MOTHER OF FLOWERING TIME 1* (*MFT1*), and *SHORT PANICLE 1* (*SP1*) in shade-grown panicles of SP [8]. Despite many physiological, biochemical, and molecular mechanisms known in SP for low-light tolerance [8,36,37,38], the epigenetic mechanisms and phenotypes controlled by miRNA-mediated PTGS for sustainable yield in naturally occurring, shade-tolerant rice varieties are not well-known [39,40]. Next-generation genomics-based small RNA sequencing has enormous advantages over the conventional cloning, sequencing, RACE, and Northern blotting-based identification of miRNA, all of which can identify miRNA on a genome-wide scale. Moreover, small RNA sequencing combined with degradome sequencing has advantages over traditional target finder tools. Degradome sequencing not only detects thousands of targets sliced by miRNA in a high-throughput manner but also provides cleavage-specific data [41]. Recent studies have also shown the unequivocal efficacy of degradome analysis in both biotic and abiotic stress research in plants [42,43,44]. In this study, to identify the miRNAs and to understand their regulation for prolonged-shade tolerance in Swarnaprabha (SP) rice, small RNA sequencing was conducted with panicles on the day of complete emergence. Degraded targets were identified using degradome sequencing in the same samples. The expression levels of the cleaved targets were confirmed using microarray and qRT-PCR. Another aim of this study was to identify the shade-tolerant phenotypic responses in SP that were controlled by miR-mediated PTGS mechanisms. This growth stage was selected to focus on miRNA regulations at the reproductive stage, which is of agronomical significance. The results of this study demonstrate new miRNA regulations for prolonged low-light tolerance (20 days) in the reproductive stage of the promising shade-tolerant rice variety Swarnaprabha.

## 2. Materials and Methods

### 2.1. Plant Growth Condition and Sample Preparation

Swarnaprabha rice seeds were imbibed on wet blotting paper and incubated at room temperature for 48 hours (h), followed by 15 days (d) of growth under white light. Seedlings were grown in pooled a pot for the next one month. Plantlets were then transferred to single plant/pot and grown until the 35th day. Plants were either grown under natural daylight (which was considered to be the sun condition) or continuous shade until the 55th day after sowing. Shade was created by covering the net house with 75% quantified cut-off agronet from B&W Agro Irrigation Co., Mumbai. Plants were grown in 12-inch pots with 8 L of soil from rice fields, mixed with urea (1 g), potassium (200 mg), and phosphate (2 g). Urea was applied thrice (i.e., during the soil preparation, in the growth stage, and before booting). Standard growth was maintained in all plants via uniform watering and insecticide spraying. Samples were collected twice from plants grown in a completely randomized block design in the net house during July–November 2018 (season 1) and July–November 2019 (season 2). The natural day light varied between a maximum of 1436.5 μmols m^−2^ s^−1^ and a minimum of 391.9 μmols m^−2^ s^−1^ in season 1 and a maximum of 1397.9 μmols m^−2^ s^−1^ and a minimum of 405.7 μmols m^−2^ s^−1^ in season 2. NISER was located at an altitude of 38 m above sea level, with a latitude/longitude of 20°09′35′′ N 85°42′26′′ E. The temperature, humidity (RH), and rainfall (RF) (as recorded by The Metrological Department situated at experimental station) presented an average 34 °C max, 23 °C min, 67% RH, and 260 mm RF in season 1; an average max of 41 °C, 21.3 °C min, 36% RH, and 163.7 mm RF in season 2; and an average 40 °C max, 19 °C min, 46.5% RH, and 116.2 mm RF in season 3. Samples for the small RNA and degradome sequencing were prepared from Swarnaprabha rice panicles grown under sun or prolonged-shade conditions on the day of complete emergence. Each sample comprised a collection of spikelets from the mother panicle of 3 individual plants in each season. Spikelets were immediately frozen in liquid nitrogen in the same light condition in which they were harvested.

### 2.2. Small RNA Library Preparation and Sequencing Analysis

For RNA isolation, 100 mg of plant tissue was homogenized with liquid nitrogen using a pestle and mortar with a lysis buffer and a TOMY smasher homogenizer. Total RNA was extracted using the Sigma Spectrum Plant Total RNA Kit (Cat# STRN50-1KT) as per the manufacturer’s protocol, including on-column DNase treatment. RNA was eluted in Nuclease-free water (Cat# AM9932, Ambion, Thermo Scientific, Waltham, MA, USA). The quantification and quality of the RNA was confirmed using Nanodrop 2000 (Thermo Scientific, Waltham, MA, USA), Qubit (Thermo Scientific, Waltham, MA, USA) and Bioanalyzer 2100 (Agilent, Santa Clara, CA, USA) with RIN ≥ 7.

For the small RNA sequencing, four libraries were prepared from two sun-grown and two shade-grown RNA extracts with the NEXTflex™ Small RNA Sample Preparation protocol (Bio Scientific Corporation, Austin, TX, USA). All procedures for small RNA sequencing were performed at Genotypic Technology Pvt. Ltd., Bangalore, India. Briefly, adapter-ligated fragments were reverse-transcribed using M-MuLV reverse transcriptase, and the thus-formed cDNA was enriched and barcoded using 17 cycles of PCR amplification. The Illumina-compatible sequencing library was quantified with a Qubit fluorometer (Thermo Fisher Scientific, Waltham, MA, USA), and its fragment size distribution was analyzed with an Agilent 2200 Tape station. The small RNA sequencing was carried out for 75 cycles on the Illumina NextSeq 550 High Output sequencing platform following the manufacturer’s instructions.

The quality of the raw data (75 bp length) was checked using FastQC (FastQC). After the elimination of low-quality bases (<q30), sequences were trimmed and filtered using sRNA workbench V3.0_ALPHA [45]. Non-coding RNA including rRNA, tRNA, snRNA, and snoRNA contamination was eliminated using Rfam [46]. An average of 1.7 million high-quality and non-redundant reads were retained for further analysis (Appendix A). Nearly 1.25 million high-quality reads, which comprised 87.81% of the *Oryza sativa* genome, were aligned, thus indicating a sound platform for further analysis. Known miRNAs were determined via a homology search of reads (≥17 nt and ≤25 nt) against *Oryza* mature miRNA sequences retrieved from the miRbase-22 [47] database using NCBI-Blast-2.2.30+ [46] at e^−4^ and non-gapped alignment. Sequences with no homology were used for novel miRNA prediction based on the proper stem–loop secondary structure using Bowtie 1.1.1 [48] and Mireap_0.22b [49] with the following major criteria: a maximum energy of ≤−18 kcal/mol [50] and a maximal mature bulge in stem–loop of ≤4. (Appendix A). The potential mature novel miRNA was selected with the most abundant read sequence that aligned with the potential precursor sequence. The expression level of novel miRNA was then specified as the sum of an ensemble of reads that aligned with the potential mature molecules, allowing for three nucleotides sliding beyond the position of the potential mature miRNA at the 5′ end. Out of the 48 novel miRs detected, 42 were retained after removing duplicates (Appendix A) and only 9 were shortlisted for secondary structure predictions based on the normalized expression count of >10. An average of 311 known miRNAs was predicted from the SP control and shade-treated small RNA samples (Appendix A) belonging to 64 different families. Target prediction was conducted with *Oryza sativa* mRNA sequences as the reference and known (copy number ≥ 5) or novel miRNAs as inputs, along with reference transcript sequences input to psRNA target tool [51]. For differential expression, absolute counts were identified and used in calculation by the DESeq tool [52]. For upregulated, downregulated, or neutrally regulated DEms, a log_2_ fold change cut-off value of ±1.0 was used.

### 2.3. Phylogenetic Analysis of MIR Families

The miRNA sequences from different obtained plant species were compared with rice miRNA sequences to check the similarity of different miRNAs with rice miRNAs. The multiple sequence alignment of different miRNAs was performed using ClustalW [53]. The phylogenetic tree was constructed using Molecular Evolutionary Genetic Analysis, MEGA 6.0 [54].

### 2.4. Degradome Sequencing, Target Identification and Analysis

After the RNA quantity and quality check, two degradome sequencing libraries were prepared using protocol from TruSeq Small RNA Sample Preparation (Illumina, San Diego, CA, USA). PolyA mRNA was isolated using 50 µg of total RNA by employing the Dynabeads Oligo dT 25 magnetic isolation kit followed by 3′ and 5′ adapter ligation. Adapter-ligated fragments were reverse-transcribed with Superscript III reverse transcriptase (Invitrogen). The thus-formed cDNA was enriched and barcoded by PCR amplification (14 cycles). The amplified library was size-selected (140–350 bp), followed by overnight gel elution and precipitation. Illumina-compatible sequencing libraries were quantified with a Qubit fluorometer (Thermo Fisher Scientific, Waltham, MA, USA), and fragment size distribution was analyzed with an Agilent 2200 Tape Station. The libraries were paired-end sequenced on an Illumina HiSeq X Ten sequencer (Illumina, San Diego, CA, USA) for 150 cycles following the manufacturer’s instructions. The data obtained from the sequencing were demultiplexed using Bcl2fastq software (v2.20, Bcl2fastq: https://sapac.support.illumina.com/sequencing/sequencing_software/bcl2fastq-conversion-software.html) (accessed on 1 January 2022), and FastQ files with raw reads of 75 bp were generated based on the unique dual barcode sequences. The sequencing quality was assessed using FastQC v0.11.8 (Fast QC). Trim Galore (https://www.bioinformatics.babraham.ac.uk/projects/trim_galore/) (accessed on 1 January 2022) was used to trim the 3′ adapter and to perform length filtering (minimum length of 16 bp). The low quality and contaminating reads were removed by performing alignment against ncRNAs (Rfam) using Bowtie [47] while meeting the following criteria to obtain final clean reads: the elimination of low-quality reads, the elimination of reads without 3′ adapters, the elimination of reads <16 bp, and the elimination of reads matching to other ncRNAs (r, t, sn, and snoRNAs). The processed FASTA sequences were analyzed using the CleaveLand pipeline [55].

To identify the miRNA-mediated cleavage sites, the known and novel miRNA sequences of all 4 samples from the small RNA sequencing were considered as input (Appendix A). The small RNA sequencing reference transcriptome and miRNA data were used for degradome data analysis. For each exact match to the sense strand of an mRNA transcript, a 26 nt long “query” mRNA sub-sequence was generated by extracting 13 nt long sequences upstream and downstream of the location of the 5′-end of the matching degradome sequence. All query sequences were aligned to each small RNA sequence using the Needle program in the EMBOSS package [56]. Alignments were then scored according to the scheme developed for plant miRNA/target pairings [57]. All alignments with scores not exceeding the user-set threshold (0.65) and having the 5′-end of the degradome sequence coincident with the 10th nucleotide of complementarity to the small RNA were retained. The hits were categorized into 5 degradome categories based on the abundance of the diagnostic cleavage tag relative to the overall profile of degradome tags matching the target. Category 0: >1 read equal to the maximum on the transcript when there is just 1 position at the maximum value; category 1: >1 read equal to the maximum on the transcript when there is >1 position at maximum value; category 2: >1 read above the average* depth but not the maximum on the transcript; category 3: >1 read but below or equal to the average* depth of coverage on the transcript; and category 4: 1 read at that position. “*” means that the average does not include all of the ‘zeroes’ for non-occupied positions within a transcript. Instead, the average is the average of all positions that have at least one read. To avoid false-positive results, targets with degradome categories between 0 and 4 were considered for further analysis.

### 2.5. Transcript Expression Analysis of miRNA and Target

The relative transcript expression of selected miRs was validated using stem–loop qRT-PCR. One gram of purified RNA was used as the starting material. cDNA synthesis was conducted according to the Agilent High Specificity 1st strand cDNA synthesis protocol. The assay reactions were performed in Strategene Mx3005P using EvaGreen relative quantification (Bio Rad) according to manufacturer’s protocol. For internal control, U6 primers were used. From the degradome analysis, the targets that were cleaved by the miRNA under both sun and shade were searched in the microarray expression data [8]. For this study, the transcript expression of Log_2_ FC (±0.25) was taken as up- or downregulated and further verified with real-time PCR. All expression levels were first normalized to their respective UBIQUITIN (OsUBQ5) expression levels. For relative expression levels and fold change (FC) calculation, the expression level under sun was used as the control. The relative transcript levels were calculated as Log_2_ fold changes. Fold changes in gene expression were calculated with the 2^−ΔΔCt^ method [58]. Each real-time datum was a mean of 2 biological replicates, each performed in triplicate. The primers of targets verified using qRT PCR are listed in Appendix A. The transcript levels determined with qRT-PCR were normalized against the transcript expression of ubiquitin.

## 3. Results

In this study, small RNA sequencing and analysis were conducted with two control libraries of sun-grown samples and two test libraries of shade-grown samples. After the removal of low-quality reads, 1,733,746 and 1,966,935 trimmed unique reads from the control libraries and 1,460,402 and 2,035,582 trimmed unique reads from the test samples were obtained from the four small RNA libraries. The cleaned reads that aligned to the genome had 83.8%, 89%, 87.1%, and 91.2% of coverage, respectively (Appendix A). A correlation plot analysis for the aligned sequences among the four libraries showed higher correlations among the two control libraries or the two test libraries than between the control and test libraries (Figure 1A). This result indicated collinearity among the control or test replicate sequences for further analysis. Genome alignment tests ranked 14,694 and 12,535 reads as uniquely aligned in the control and test groups, respectively, and ranked 19,025 reads as common to the control and test (Figure 1B). This result indicated a stronger overlap of reads and a better platform for further analysis. The length distribution analysis of sequences showed that the majority of our libraries were in the 24 nt size class, indicating suitability for further comparative analysis (Figure 1C). The 24 nt sequences accounted for 37% and 36% of the total reads of the control replicates and 27% and 36% of the total reads of the test replicates. The higher read count of 24 nt sequences in the second shade replicate (shade_2) again supported the use of this sound platform for further analyses.

### 3.1. Analysis of Known and Novel miRNAs

Known miR length ranged between 19 and 25 nt, with the size of 21 nt being the largest category (51.3%). The novel miRNA size ranged between 20 and 23 nt, and each size of miR had a homogenous proportion (Figure 1D). A lower amount of 5′ uridine of the miR diminished its stability by preferentially loading it onto an AGO complex. The frequency of the first nucleotide bias of the mature miRNA showed distinct differences among the known and novel miRs. More than 40% of the known miRs had a uridine at the 5′ end, and nearly 50% of the novel miRs preferentially started with a cytidine (Figure 1E). This indicated that the known miRs had a shorter half-life than the novel miRs, as adenine–uridine (AU)-rich elements (AREs) contribute to a shorter half-life and the absence of AU or UA dinucleotide elements may confer miRNA stability and lengthen the half-life. The percentage base composition of the known miRs revealed that the 23 nt size had only 10.8% of cytidine, indicating lesser stability (Figure 1F), while the percentage base composition of the novel miRs was relatively homogenous, with ~25% for each base (Figure 1G). Consistently, the 21 nt known miRs were enriched with uridine as their first base, and 23 nt miRs had adenine in nearly 50% of cases (Figure 1H). The 20 and 21 nt novel miRs had a cytidine as their first base in nearly 70% of instances, indicating more stable forms than the known miRs (Figure 1I). The formations of a stable hairpin structure, internal stem–loops, and mis-paired RNA ‘bulges’ are highly conserved and necessary requisites for different pre-miRs across species. Secondary structure prediction for the novel miRs with the help of an RNA-fold server [39] (http://rna.tbi.univie.ac.at/cgi-bin/RNAWebSuite/RNAfold.cgi) (accessed on 1 January 2022) was conducted. From the selected novel miRs (Appendix A), a further 5 distinct novel miRs were shortlisted on the basis of the absence of uridine > 8, as there are no reported miRNAs in plants containing stretches of A7, C8, G6, or T7 [15]; the presence of secondary and centroid structures; clear 5′ and 3′ arms; miRNA precursor regions; mature miRNAs (marked with color) and secondary hairpin structures, and 5′ ends loaded with either an adenine or uridine (Appendix A). A 2 nt overhang at the 3′ end serves as a recognition site for the Dicer, which cleaves the pre-miRNA ~20 nt upstream of the 2 nt 3′ overhang recognition site to produce a miRNA duplex intermediate for secondary processing. Such an overhang at the 3′ end was clearly observed in our selected novel miRs.

### 3.2. Similarity Analysis of Rice MIR Families with Other Species

A total of 412 miRs were retained for analysis after the removal of duplicates, which were identified based on similarity in the miRbase (Appendix A). The family distribution analysis of known miRNAs revealed MIR812 as the most abundant family, followed by MIR2118 and MIR396 (Appendix A). The known miRNAs presented similarity with 27 genera and 31 species (Appendix A). Two species of Arabidopsis (*A. thaliana* and *A. lyrata,* including 13 different miRs) showed similarity to the highest number of 11 MIR families, followed by 9 MIR families that showed similarity to 8 different miRs of *Medicago truncatula* (Appendix A). Furthermore, the MIR144 formed a subgroup with *Arabidopsis thaliana*. The MIR family MIR5067 showed the highest similarities with different species, i.e., 24 miRs (including *T. aestivum*, *H. vulgare*, *A. tauschii,* and *B. distachyon*) (Figure 2A). According to the phylogenetic analysis (Figure 2A), *A. thaliana* (10), *B. distachyon* (7), *T. aestivum* (7), and *Z. mays* (7) shared maximum identity with rice miRNAs and were among the closest species, and the *Selaginella melandasy* species was the most related ancestor in relation to miR phylogeny. MIR529 was clustered with *Physcomitrella patens*, indicating the conserved phylogenetic relationship of the rice miRNAs with this species. The similarity analysis of the novel miRs for orthologues in other plants revealed only one hit, i.e., mireap-m0014-5p with the 5′TGTATGGCTCTGATACCAGCTCT3′ sequence, the showed 100% identity (E value 0.002) with the *Rehmannia glutinosa miR5139* belonging to miRNA family of *ap-m0014-5p* (data not shown). However, considering the stability and no homology, we suggest that new species-specific miRNAs have recently evolved and been expressed at lower levels [59]. A list of MIR families and their rice miRNAs is presented in Appendix A.

The phylogenetic analysis demonstrated a clear divergence of different miRNAs found under the sun and stress conditions (Figure 2B). Based on the similarity of the sequences, miRNAs were grouped into two major clusters from 31 different species in the sun and shade conditions (Figure 2B). In both sun and shade conditions, cluster I had 4 subclusters. While under sun cluster II has 3 subclusters, under shade conditions, cluster II had 5 subclusters. The maximum numbers of miRs were grouped in clusters I and II under the sun and shade conditions, respectively. Most of the rice miRNAs from clusters I and II showed similarity with miRNAs of other species under the sun and shade conditions, respectively. Among MIRNA families, MIR1082 in *S. moellendo* and MIRNAs 529, 537, 899, 902, 1023, and 1027 in *P. Patens* were found to be ancient and evolutionarily conserved with microRNAs of the rice genome. The evolutionary divergence predictions drawn from this result indicated that six rice miRNAs (i.e., MIR529, MIR537, MIR899, MIR902, MIR1023, and MIR1027) branched together with *S. moellendo*, thus suggesting they were ancient and evolutionally conserved miRNAs.

### 3.3. Differential Expression Analysis of the Known and Novel miRNA

Differential Expression analysis of the known and novel miRNAs: The differential expression analysis of the miRs from sun and shade-grown samples revealed 323 differentially expressed (log_2_ FC ± 1.0) miRs (DEms), whereas 47 and 42 miRs were unique to the sun- and shade-grown samples, respectively (Appendix A) (Figure 3A).

A total of 263 known miRs with *p* ≤ 0.9 were filtered. Among the DEms, 40 were upregulated, 54 were downregulated (Appendix A), and 169 were neutrally regulated (Appendix A). From the 42 novel miRs, 24 and 15 miRs were uniquely expressed in the sun and shade-grown samples, respectively (Appendix A), and 3 miRs were differentially expressed (Appendix A). The heat-map analysis of the known DEms conducted for top 20 up- and downregulated miRs (Figure 3B) under shade showed the maximum contrasting regulation of FC among the DEms. These samples included *miR5146*, *miR399i* and *miR168b*, all with the maximum number of targets (Appendix A). The volcano plot analysis of the known DEms (Figure 3C) showed that the maximum number of DEms were placed below *p* = 0.6 and were thus suitable for further study. Among the DEms, miR2875 had the highest abundance of +3.3 FC (Figure 3D) and miR1850.2 had the least abundance of −3.2 FC (Figure 3E). The phylogenetic analysis of the DEms from the sun and shade conditions showed drastically different clustering patterns. *miR159*, *miR393*, *miR399*, *miR444*, and *miR2275* were branched together in a single clade (clade I), whereas *miR414* and *miR529* diverged as another clade (clade II). *miR529* was clustered in clade I under shade and in clade II under sun. Conversely, *miR399* was clustered in clade I under sun and in clade II under shade (Figure 2B).

### 3.4. miRNA Target Identification Using Degradome Sequencing

Degradome sequencing was adopted to obtain multiple miRNA cleavage sites on a genome-wide scale using the high-throughput sequencing-based approach. A simplified workflow of data analysis and candidate selection is presented in Appendix A. A total of 62.58 million raw reads were generated from sequencing, including 35,664,885 reads from control and 26,914,530 reads from the shade libraries (Appendix A). After removing reads >16 bp and trimming adapters, 31,460,618 cleaned reads from the sun condition and 20,549,308 reads from the shade condition were obtained. Alignment to the rice genome resulted in 29.96 million quality and non-redundant reads (including 17,040,807 reads from the sun condition and 12,923,499 reads from the shade condition) for target identification using degradome sequencing (Appendix A). We identified 4092 sites for known miRNA-mediated cleavage and 430 sites for novel miRNA-mediated cleavage in the sun degradome sequences. Similarly, 1488 sites were identified for known miRNA-mediated cleavage and 178 sites for novel miRNA-mediated cleavage in the shade degradome sequences (Appendix A). These cleavage sites were from 2672 and 1053 transcripts from the degradome targets of sun and shade libraries, respectively, under the cleavage category (0–4) after filtering, as mentioned in the Section 2.4 (Appendix A). However, 1401 and 448 targets were found to be targeted by the miRs of the sun and the shade libraries, respectively. Similarly, in case of novel miRs, 386 and 163 transcripts were found in the degradome of the sun and shade libraries, respectively (Appendix A). Furthermore, 37 and 10 transcripts were commonly targeted by the miRs and the degradome target libraries, respectively. These results indicated that nearly 6.9% and 1.1% of the targets were sliced by the known and novel miRs under sun, respectively, whereas 2.9% and 0.4% of the targets were sliced by the known and novel miRs in the shade library, respectively. The cleavage of the targets was confirmed by the study of 5′ end frequency, degradome category, and amplitude from their respective target plots (t-plot) (Figure 4). Ultimately, 50 miRs were found to be DEms under both the sun and shade conditions, including 13 up- and 37 downregulated miRs (Appendix A). These results indicated that the proportion of downregulated miRs was nearly three-times higher than the upregulated miRs under shade.

### 3.5. Expression Analysis of miRNA Targets Using Microarray and qRT-PCR

The expression levels of the targets were verified using the microarray genome-wide expression analysis from the panicle samples of SP grown in the sun compared to the similar shade treatment [8]. The targets of the DEms from the degradome sequencing analysis were searched in microarray data, and a total of 191 transcripts were obtained. After filtering the log_2_ fold-change cut-off value of ±0.25 for up- or downregulation, totals of 51 and 40 transcripts were obtained as DEGs with up- and down regulation, respectively. The opposite expression levels between miRNAs and their targets were observed in 8 miR–target pairs of up-DEms and 34 miR–target pairs of down-DEms (Appendix A). The top 3 up- and 13 down-DEms, as well as their respective pairs, are presented in Figure 5A. Some miRs targeted single transcripts, and some cleaved multiple ones. The relative expression levels of 6 miRs and their 11 targets were verified using qRT-PCR (Figure 5B). Although very few of the expression levels obtained from qRT-PCR were different from those obtained with the sequencing analyses, most of their expression patterns and abundances were consistent with those from the miRNA sequencing and microarray data.

### 3.6. Pathway Distribution Analysis of Predicted Targets of miRNA and the Degradome Targets

The pathway analysis of the predicted targets of miRNA transcripts using the UniProt database and the RAP-DB and GO annotations presented an overview of the biological function of the targets of the miRs selected for shade tolerance. The miRNA–target prediction analysis resulted in 35,672 transcripts, with 20,215 and 15,457 transcripts being the targets of the miRs from the sun and shade libraries, respectively. Similarly, in the case of novel miRs, 3405 and 2367 transcripts were found to be targets of novel miRs from the sun and shade libraries, respectively. The pathway enrichment analysis of the predicted targets of miRs (Appendix A) demonstrated the appearance of categories of cytoskeleton organization in the biological process, cytoskeletons in the cellular component and slightly reduced MAP kinase activity, structural constituents of cytoskeletons, and cellulose synthase activity in molecular function in the shade-treated samples. These results indicated that there was specific activity girdling cell wall, cytoskeleton, and cellulose synthesis due to the decrease in incident light in the shade-treated samples. Moreover, a decreased activity of nutrient and stress compound metabolism was also noticeable in targets of the shade-treated samples. The predicted targets of miRs from the small RNA sequencing served as the inputs to filter the targets cleaved specifically during prolonged shade in the degradome sequencing data. Furthermore, out of the total 87 targets, 30 targets were filtered after being found in the common degradome of sun and shade libraries (Appendix A). The pathway analysis of the degradome targets common between sun and shade condition showed that the transcripts related to transcription factors, DNA and RNA binding, and transcriptional and post-transcriptional events comprised the maximum of 13.6%, followed by cell-wall- and membrane-related transcripts that comprised 11% of the degradome transcripts (Figure 6A).

The pathway analysis of the targets of DEms presented drastic differences in the category of targets according to the function and pathway in they were involved (Figure 6B). The upregulated miRs mostly targeted (Appendix A) the transcripts of cell wall and membrane proteins, biotic and abiotic stress signaling, and hormone signaling pathways. The downregulated miRs mostly targeted (Appendix A) the transcripts of photosynthesis, carbon metabolism, sugar metabolism, energy metabolism, and amino acid and protein metabolism. The percentages of these transcripts in the downregulated and upregulated DEms were different, apart from the primarily affected category of transcription factors in both cases. The downregulation of the cell wall and membrane transcripts was probably caused by the specific regulation of 4 up-DEms, namely *miR2094-3p*, *miR414*, *miR6245*, and *miR2118* (Appendix A) (Figure 4E, Figure 4F, Figure 4G, and Figure 4H, respectively). The photosynthesis, carbon metabolism, and sugar metabolism transcripts that were upregulated were also targeted by the down-DEm *miR5493*, which targeted the maximum number transcripts in this category (Figure 5A). The highest number of targets among the energy metabolism category were upregulated by the down-DEm *miR399i* (Figure 4A). Additionally, the down-DEm *miR5146* showed maximum target binding in the amino acid and protein metabolism transcripts (Appendix A) (Figure 4C), which were upregulated. Transcripts of heat shock proteins and chaperones, secretory pathway, and calcium signaling transcripts were shown to be the exclusive targets of the down-DEms (Appendix A). These results indicated that different miRs are involved in preforming different functions during prolonged shade in SP.

### 3.7. miRNA Regulation According to Functional Properties of the Targets

miRNA regulation according to the function of the targets included transcripts of transcription factors, cell walls, membrane dynamics or hormone signaling, photosynthesis and carbon metabolism transcripts, shade tolerance, yield, abiotic stress, light signaling, and flowering. The specific binding of miRs and their targets selected in this study was confirmed by analyzing their alignment [60], as presented in Figure 7. Novel regulation due to neutrally regulated and uniquely expressed miRNAs was also clearly observed due to the differential specificities for their targets (Figure 8). Their stable binding was re-affirmed by analyzing the free energy change, Allen score, secondary structure, and *p*-values presented in Table 1. Again, the cleavage of target by the miR was supported by the T-plot degradome categories presented in Figure 4. The molecular analysis of the relative expressions of some selected targets and their respective miRs is presented in Figure 9 to support the NGS and microarray results.

#### 3.7.1. miRNA Regulation of Transcription Factors

The pathway analysis of the commonly degraded transcripts revealed that transcripts related to DNA and RNA binding, transcriptional events, post-transcriptional events, and transcription factors (TF) comprised a maximum of 13.6%. These included targets of both the up- and down-DEms in equal proportion (Figure 6B). The TFs were categorized into five types (i.e., WRKY type, AP2/ERF type, B3 type, NAC type, and SBP type) reported as specific to plants [59]. Half of the total TFs obtained in the degraded transcripts (8/16) were of the WRKY type, including MYB, bHLH, SANT, and Tify domain-containing proteins (Appendix A). The WRKY group of TFs is well-known in biotic and abiotic stress responses [59]. The second largest category of TFs in the degraded transcripts comprised the AP2/ERF type (25%; 4/16), including ERF, EIL, and EIF, followed by 18.7% (3/16) of the B3-type TFs including bZIP, homeodomain, HD-ZIP, and LSM. These results indicated that miR-controlled regulations for prolonged-shade tolerance mostly focus on transcriptional activation, deactivation, or DNA- and RNA-binding events. The miRs targeting the maximum number of transcripts in this category were *miR168b*, *miR5542*, *miR5146*, and *miR399i*. The *miR5493*-mediated upregulation of *Os06g0592500*, an ethylene-responsive transcriptional coactivator (ERTC) (Figure 5A and Figure 9A; Appendix A) that is induced during heat stress, was notable [61]. Possibly, *miR5493* cleaved *ERTC* at 249 nt (Appendix A) under control conditions, which was then released due to the decreased abundance of *miR5493* under shade.

#### 3.7.2. Regulation by Uniquely Expressed and Neutrally Regulated miRNAs

Most of the studies regarding miRNA regulation have focused on regulation by miRNAs that are differentially expressed between the control and the treatment libraries. However, we observed novel regulation by the uniquely expressed miRNA from the sun or shade degradome libraries caused by the differential expression of their targets. The unique expression of miRNAs in the shade samples with targets that were differentially downregulated under the shade condition indicated the possible downregulation of the target due to the unique expression of the corresponding miRNAs. In contrast, for the uniquely expressed miRNAs in the sun condition, the upregulation of the corresponding target under shade might take place due to the absence of miRNAs leading to an increase in the target abundance under shade. We identified 3 such transcripts being downregulated and 15 such upregulated transcripts under shade (Figure 8) (Appendix A). The 83 targets of novel miRs uniquely expressed under sun (Appendix A) and the 23 targets of novel miRs uniquely expressed under shade (Appendix A), which were filtered for having selection criteria of degradome category and Log_2_FC were also analyzed under this study.

The unique expression of *miR160b-3p* under shade (Figure 8) resulted in the cleavage of *Os04t0104900-01* (*OsCOMTL2*, a methyltransferase) at CS 230 nt (Figure 4L) and its downregulation (−1.8-fold) (Appendix A). Among the uniquely expressed miRs under sun, *miR169-p* sliced its target *Os01t0188400-01* (*OsChlME*, NADP-dependent malic enzyme) at CS 1635 nt (Appendix A) (Figure 8). The absence of miR169-p under shade probably led to the increased abundance of *OsChlME* and thus its differential upregulation to 1.24-fold under shade. All members of the miR169 family were significantly upregulated in the phytochrome B (*phyB*) mutant [15]. Our results regarding the unique expression of *miR169p* supported the previous observation of the *miR169* family under PhyB and light signaling. A novel miRNA (with a 5′*UCGUGCCGGCGGGGGCCGGGCU*3′ sequence) found uniquely in the sun condition cleaved its target *OsDELLA* (*Os03t0707600-01*, the key repressor of gibberellin signaling) at CS 166 nt, resulting in differential upregulation (log_2_FC 0.62) under shade (Appendix A). This result indicated the probable absence of this novel *miR*, resulting in an increased abundance of *OsDELLA* under shade (Figure 8). Enolase 2 (*Os03t0248600-01*, a carbon metabolism transcript related to chloroplastic glycolysis) [62] was found to be upregulated (Log_2_FC 0.7) with CS 1442 nt by a uniquely expressed novel miR with a 5′*AAAGUAUCAAGUUUAAAUUCAU*3′ sequence (Figure 4M), which was expressed under sun (Appendix A). It was suggested that the absence of *AAAGUAUCAAGUUUAAAUUCAU* under shade probably led to an increased abundance of *OsENO2-1* (Figure 8).

To identify the regulation controlled by the neutrally regulated miRNA, we hypothesized that despite not being up- or downregulated, they might control the differential abundance due to differential binding with their target in the sun or shade. We found 14 such miRs mediating the differential abundance of their targets (Appendix A). The downregulation of protein phosphatase 2C (*OsPP2C*, *Os03t0761100-02*) to −1.53-fold (Figure 9B) and cleavage due to the binding of the *miR399b* (Log_2_ FC = 0.66) (Figure 4N) are notable. Rice PP2C is known for its large subfamilies, function stress tolerance, and ABA-mediated signaling pathway [63]. A cyclin F-box-containing protein (*OsFbox6*, *Os01t0281000-01*) could be upregulated (Figure 9B) due to binding of its target *miR1439* (Log_2_ FC = −0.38), indicating degradation events specific to the shade condition (Appendix A). Out of the total five TIFY-domain-containing proteins that were identified to be differentially regulated in SP panicles [8], only *TIFY11D* was found to be upregulated (Figure 9B) and targeted by a neutrally regulated *miR5810* (Appendix A). miR5180 cleaved *OsTIFY11D* at 699 nt (Figure 7B). The overexpression of *OsTIFY11D* (*Os10t0392400-01*, a jasmonate signaling gene) resulted in tolerance to various abiotic stresses such as salt and dehydration [64]. However, our finding of the upregulation of *TIFY11D* could indicate the involvement of *miR5810* and jasmonate signaling in SP panicles under prolonged shade conditions. The Lonely Guy 1 (*OsLOG1*, *Os01t0588900-01*, a cytokinin-activating enzyme known for controlling panicle size) [65] was sliced at 483 nt (Figure 4J and Figure 7E) by *miR5144-5p*, which was categorized as a neutrally regulated miR in our data. Despite the *miR5144-5p* being neutrally regulated, it may contribute to differential binding with its target and play a significant role in the upregulation of *LOG1*, resulting in a longer panicle length in SP under shade. *miR5144-5p* was also found to target *OsWD40-24* (*Os01t0725800-01*, *SPA3-4-like*) (CS 446 nt; Figure 4K), which was upregulated in the shade-grown SP panicles (Figure 9A and Appendix A). In addition to being repressors of photomorphogenesis, *SPA2* and other members of the SPA family have also been shown to regulate light-induced stomatal closure, mesophyll photosynthesis, and sucrose breakdown [66]. Hence, it could be suggested that shade-induced differential binding of *miR5144-5p* followed by the targeted cleavage of *OsWD40-24* could play a regulatory role in sustained yield under shade in SP.

#### 3.7.3. miRNA Regulation of Transcripts of Cell Wall, Membrane Dynamics or Hormone Signaling

The up-DEms were found to specifically control the downregulation of cell-wall-related transcripts (Figure 6B). The analysis of cell wall and membrane transcripts (Appendix A) showed that miR414 mediated the downregulation of Os06t0561200-01 (a potassium/proton antiporter) (CS 142 nt; Figure 4F), Os05t0146100-01 (PDZ/DHR/GLGF domain-containing protein) (CS 1863 nt), and Os10t0503800-01 (a remorin) (CS 744 nt) (Appendix A; Figure 9B), and miR6245 mediated the downregulation of Os07t0551600-01 (OsCslF9, cellulose synthase-like,) (CS 1723 nt), which was specifically induced under prolonged shade (Figure 4B, Figure 5A, Figure 7A and Figure 9A; Appendix A). The functional properties of all these target genes, which encode either an antiporter, an ion channel [67], or a membrane protein [68], indicated reduced transport across the cell wall, membrane, or ion channels, respectively. Furthermore, three miR–target pairs (imiR5075-OsDREPP2, miR529a-OsDjA5, and miR444.2-OsEREBP96) (Appendix A) with downregulated miRNAs and their targets (CS 369 nt, 1426 nt, and 575 nt, respectively) showed upregulation. Notably, EREBP96 functions as a transcription factor in ethylene signaling, J-proteins are members of co-chaperones that assist HSP70 [69], and OsDREPP2 is a developmentally regulated plasma membrane protein. These results confirmed that miRNA regulation for shade-tolerance could concern membrane dynamics, hormone signaling and HSPs.

#### 3.7.4. miR Regulation of Photosynthesis and Carbon Metabolism Transcripts

Transcripts of photosynthesis, carbon metabolism, and sugar homeostasis were targeted by a higher proportion of down-DEms. Moreover, down-DEms specifically targeted energy metabolism and amino acid and protein metabolism (Figure 6B). Among photosynthesis-related transcripts (Appendix A), the upregulation of fructose biphosphate (*OsFBA*, *Os05t0402700-01*) (Figure 9A) being sliced by *miR2275b* at CS 341 nt was noticeable (Figure 7C). Among the targets of *miR5493*, which sliced the maximum number of photosynthesis-related, carbon metabolism, and sugar homeostasis transcripts, the upregulation of *Os01t0226600-01* (*OsSLAC*, C4-dicarboxylate malic acid transporter) with CS 657 nt (Figure 4D) and *BRITTLE 1-1* (*OsBT1-1*, *Os02t0202400-01*) (Figure 9A) with CS 1038 nt (Appendix A) were significant. *miR5487*, a down-DEm, was found to slice a glycoside hydrolase, *Os02t0771700-01* (*OsGns9*, β-1,3 Glucanase) (CS 878 nt; Figure 7D and Appendix A), which was upregulated under shade (Figure 9A).

#### 3.7.5. miRNA Regulation of Transcripts of Abiotic Stress, Light Signaling, and Shade Tolerance

Several studies have shown the involvement of miRs in abiotic stress signaling, light signaling, and shade-avoidance response (SAR). We identified the targets of five of such miRs (*miR156*, *miR393*, *miR172*, *miR319*, *miR530*, and *miR169*) in our degradome data and verified their expression through microarray expression analysis (Appendix A). These miRs were cut out of the DEm category due to log_2_FC and *p*-value cut-offs. Actions of *miR156* and *miR172* are required for the determination of juvenile identity and transition from vegetative stage to reproductive stage [31]. *miR156i* was found to be downregulated in our data. The upregulation of *miR172b* (Figure 9B) with the downregulation of its target *OsbHLH153* (*Os10t0167300-02*) and cleavage at 313 nt (Figure 4O, Figure 7F and Figure 9B) was observed (Appendix A). *bHLH153*, a brassinosteroid signaling pathway gene, is known to regulate flag leaf angle in rice [70]. This result indicates that *miR172b* may regulate flag leaf angle by lowering the expression level of *bHLH153* and the hyponasty of leaves, which is also a discernible phenotype under shade. In the amino acid and protein metabolism category, the down-DEm *miR168b*, whose target is *Os11t0255300-01* (a cysteine endo-protease (*OsCP1*)), was cleaved at 1044 nt (Figure 7G and Appendix A), resulting in its upregulation (Figure 9A). *OsCP1* has been shown to encode a papain family cysteine protease and play a significant role in pollen development. Though *miR530-3p* was found to be significantly upregulated in the phyB mutant [15], it was found to be neutrally regulated in our data (Appendix A). Another miRNA, *miR319*, was shown to help gain cold tolerance and chilling acclimation in rice [71], and its overexpression was found to impact leaf and floral organ development; it was neutrally regulated in our data. *miR393*, which has been shown to be involved in various abiotic stress response including salinity and cold [72], was upregulated in our data. However, its targets were not related to the shade-tolerance response and are therefore not discussed in detail (Appendix A). The regulation pattern (FC) of the *DE-ETIOLATED1* gene (*Os01t0104600-01*, *OsDET1*), involved in light signaling, was found to be contrasting with the relative abundance of its corresponding *miR168b* (Appendix A and Figure 9A). *OsDET1* was found to be sliced by *miR168b* at 366 nt (Appendix A). *OsDET1*, a homologue of Arabidopsis *DET1*, was shown to be involved in ABA biosynthesis in rice [73].

## 4. Discussion

Naturally occurring rice cultivars have tremendous potential to adapt to abiotic stresses due to the unique alleles and mechanisms they harbor that provide abiotic stress tolerance. Swarnaprabha is a well-known shade-tolerant rice cultivar, and it is thus used as a preferable resistant check in low-light tolerance studies [3,8]. Despite considerable research on SAR and its regulation at the molecular level [8,36,37,38], the involvement of miRNAs in controlling SAR and sustainable yield in shade-tolerant genotypes is not fully known. Next-generation genomics approaches of small-RNA combined with degradome sequencing has confirmed the identification of known and novel miRNAs, as well as their targets, in a high-throughput manner.

Here, we have presented a glimpse of miRNA regulation in prolonged low-light stress in SP rice accessed through genome-wide, small RNA sequencing combined with analyses of cleaved targets using degradome sequencing and transcript expression validation using microarray analysis and qRT-PCR (Appendix A). The observation of higher similarities between the datasets of the two control replicates or the two test replicates than the opposite confirmed the higher confidence of a better platform for genomic data analyses. Differential expression analysis showed 263 DEms, of which only 35% (94/263) were up- or downregulated and 65% (169/263) were neutrally regulated under prolonged shade. On the other hand, 26.8% of the miRs were uniquely expressed either in the sun or shade conditions. This scenario indicated that neutral or uniquely expressed miRs might have equally important roles as those of DEms through the differential regulation of their targets, probably due to their differential binding affinities. Hence, we analyzed the expressions of the targets cleaved by the DEms, as well as the neutrally or uniquely expressed miRs. As the increased abundance of miRs is well-known to silence their targets, only the DEm–target pairs with contrasting relative abundances were shortlisted. In our study, the cleavage of only 6.9% and 2.9% of the total miR targets under the sun and shade conditions, respectively, indicated that for shade tolerance, miR-mediated cleavage occupies only a minor proportion of the total miR regulation.

The length distribution and frequency of the first nucleotides of the known and novel miRs indicated that known miRs had shorter half-lives and were less stable than the novel miRs. This result indicated a more functional counterpart for the known miRs. MIR family analysis revealed that the highest similarity was found for *Arabidopsis* (*A. thaliana* and *A. lyrata*). Though *A. thaliana* and *A. lyrata* bore the highest similarities with a maximum of 11 MIR families, miRs of the MIR5067 family had the widest similarity to maximum of 24 different species. Hence, *Arabidopsis* miRs could be suggested as the closest to rice MIR based on sequence similarity. The study of clustering patterns and similarities of low-light stress-responsive miRs (i.e., *miR159*, *miR393*, *miR399*, *miR444*, *miR2275*, *miR414*, and *miR529*) isolated from this study showed different clustering patterns in response to shade, indicating changes in the functionalities of miRs under low-light stress.

The predicted targets of the known miRs from shade libraries showed an average 20% reduction in most of the GO terms analyzed in comparison to those from the sun libraries, indicating reductions in miRNA-regulated responses under decreased light. The pathway analysis of miRNA targets showed a reduced activity of cytoskeleton and cellulose synthase in the shade-treated samples. This decrease could have been due to the downregulation of the cell wall and cellulose synthase transcripts. This result was prominently observed from the pathway analysis of degraded transcripts, which were further categorized according to targets of up- or down-DEms. The up-DEms specifically controlled the downregulation of cell-wall-related transcripts. Similar results were observed for the nutrient and stress compound metabolism transcripts, which were specifically downregulated by the up-DEms. Remorin proteins are also known to play crucial roles in plant growth, development, signal transduction, and stress responses [68]. The *miR414*-mediated downregulation of *Os10t0503800-01* (remorin, *OsREM1.2*) might be crucial for prolonged-shade tolerance in SP. Decreased cellulose synthase (*OsCslF9*) levels indicated the decreased mechanical strength of the stem, which is a well-known effect under shade [74] that may be mediated by the upregulation of *miR6245* in SP. Nearly 3/4th of the miRs whose targets were cleaved under both sun and shade (37/50) were downregulated. This result indicated that miRNA regulation for shade-tolerance predominately comprises the deactivation of the miR itself, leading to the upregulation of their targets. The down-DEms targeted higher proportions of transcripts of photosynthesis, carbon metabolism, sugar homeostasis, energy metabolism, and amino acid and protein metabolism. The *miR5493*-mediated upregulation of *Os01t0226600-01* (*OsSLAC*) may be linked to panicle size and grain yield in SP under prolonged shade, as reports have shown that an aluminum-activated malic acid transporter localized to plasma membrane could maintain panicle size and grain yield by mediating malic acid transport [75]. The downregulation of *miR5493* also resulted in the increased expression of ADP-glucose transporter (*BRITTLE1-1*, *Os02t0202400-01*). BRITTLE1-1 has been shown to be responsible for grain formation by controlling starch synthesis [76], which supports the possible role of *miR5493* in the regulation of photosynthesis-related transcripts for sustainable yield under shade in SP. Glycoside hydrolase is primarily known to hydrolyze glycosidic linkage to release sugars, hence making a major contribution to degrading biomass [77]. Additionally, GH has been shown to have important roles in pollen development, seed germination, cold response, and plasmodesmata signaling [78]. The *miR5487*-mediated upregulation of *OsGns9* might be required to meet sugar demands due to decreased photosynthesis under low light or could be associated with the above-mentioned responses in SP. Based on our data, it could be suggested that the *miR168b*-mediated downregulation of *OsCP1* may be crucial for proper pollen development under prolonged shade in SP.

We found several miRs in our study that are known to be involved in abiotic stress responses such as cold, drought, and light signaling (i.e., *miR156*, *miR172*, *miR319*, *miR393*, *miR530*, and *miR169*). The relative expressions of these miRs and their corresponding targets may also impact shade-tolerance in SP. The observation of the *Os04t0395600-01* (transport inhibitor response 1-like protein) being the target of *miR393* was supported by similar report on Medicago truncatula [79]. Four forms of *miR169* (*miR169a*, *miR169d*, *miR169p*, and *miE169r-5p*) were found in differential abundance under sun or shade. Though *miR169a* and *miR169d* were differentially downregulated under shade, *miR169p* and *miR169r-5p* were found uniquely under sun. We have demonstrated in this study that even the neutrally regulated and uniquely expressed miRs could contribute to the shade-tolerance response in SP by altering the expression of its targets due to differential binding affinities. In this category, the regulation of *miR5144-5p*, *miR160b-3p*, *miR169p*, *miR399b*, and *miR1439* through the differential expression of their targets was noticeable. Similar kinds of the differential expression of miRNA specific to root or shoot during short or prolonged heat [80] support our observations of the differential expression of targets of neutrally regulated and uniquely expressed miRs.

Several genes known to be involved in low-light, hormone signaling, and sustainable yield conditions were verified for their relative expression and cleavage by miRNAs in this study. The miR–target pairs with contrasting expression patterns and significant fold changes and *p*-values included genes such as *OsEREBP96*, Lonely Guy 1 (*LOG1*, *Os01t0588900-01*), *OsWD40-24* (*Os01t0725800-01*, *SPA3-4-like*), and *OsTIFY11D*. An SP shade-tolerance response for up to 5 days of low-light was shown to be associated with the upregulation of the sedoheptulose 1,7 bi-phosphate (SBPase) and fructose-1,6-bisphosphatase (FBPase) components of the PSII complex: oxygen evolving enhancer proteins (*OsOEE1* and *OsOEE2*) [81]. However, it is notable that we did not find any miR that regulates the expression of these genes under prolonged low-light conditions in our study. Though we observed the upregulation of *OsOEE3* (Log_2_ FC 1.0) in our previous study [8], SBPase was found to be neutrally regulated in the present study (Appendix A). This could be explained by the report of Li et al. (2020) [82], where it was shown that expression of RuBP-regeneration enzymes such as SPBase and FBPase were drastically decreased after transient low-light exposure. Hence, under prolonged low-light conditions, SBPase expression was not significantly affected in our study. Despite the neutral expression of SBPase, we observed an upregulation of the *miR2275b* target *OsFBA Aldolase*, a nonregulated enzyme that is known to catalyze the formation of SBP and FBP, increase photosynthetic carbon flux by increasing RUBP regeneration [83], and promote gibberellin-mediated root growth in rice [84]. Hence, we suggested that the *miR2275b*-mediated upregulation of *OsFBA* aldolase may control increased root length, which is also a distinct phenotype under low R:FR in SP [8]. The increased expression of *OsENO2-1* due to the unique expression of the novel miRs (5′*AAAGUAUCAAGUUUAAAUUCAU*3′) indicated its involvement in the regulation of grain size and weight [85] through chorismite-dependent secondary metabolism and cytokinin content [86].

## 5. Conclusions

Our study revealed a complex network of miRNA regulation under prolonged shade comprising the maximal self-downregulation of miRNAs that leads to the upregulation of their targets. Through phylogenetic analysis, the similarity of rice MIR orthologues to other plants and clustering patterns of shade-responsible miRNAs could be deduced. This study has shown that for prolonged-shade-tolerance response in Swarnaprabha, miRNA regulates the upregulation of transcripts related to photosynthesis, carbon and sugar metabolism, energy metabolism, amino acid metabolism, and the downregulation of transcripts related to the cell wall (membrane transcripts). Under prolonged shade, miRNA prominently regulates activities girdling the cell wall; transport across membranes; ion channels; cellulose synthesis; secondary metabolism to control responses such as decreased mechanical stem strength, pollen development, panicle development, panicle number, and endosperm and grain development; and meeting sugar demands. Uniquely expressed and neutrally regulated miRs were also shown to contribute to shade-tolerance by altering the differential expression of their targets. We newly identified 16 miRs with 21 target pairs with contrasting regulation in this study of prolonged-shade tolerance (Table 1), including *miR5493*, *miR5144*, *miR5493*, *miR6245*, *miR5487*, *miR168b*, *miR172b*, and *miR168b* with their targets *OsSLAC*, *OsLOG1*, *OsBRITTLE1-1*, *OsCsIF9*, *OsGns9*, *OsbHLH153*, *OsCP1*, and *OsDET1*, respectively. The 16 miRs with 21 target pairs identified in this study could functionally validated with transient expression assays and for their phenotypes related to prolonged low-light stress in specific knockout or overexpression experiments. The newly identified miRNA regulations are useful for studies of sustainable crop yield under prolonged low-light stress.

## Figures and Tables

**Figure 1 biology-11-00798-f001:**
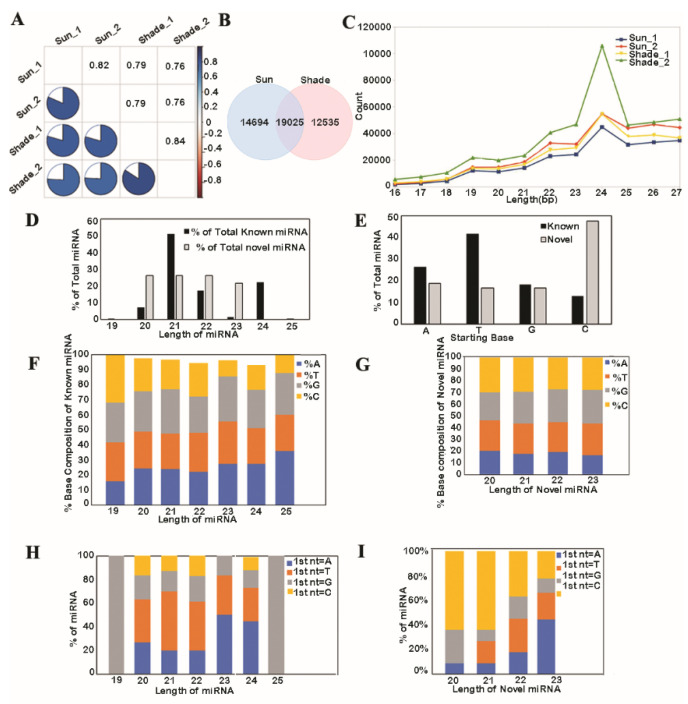
Analysis of known and novel miRNAs expressed under prolonged shade in SP. (**A**) Correlation plot analysis for the aligned sequences among the 4 libraries. (**B**) Venn diagram representation of control and test replicates’ reads that were aligned to the genome. (**C**) Length distribution of known miRNA from the 4 small RNA libraries. (**D**) Length and percentage of known and novel miRNAs. (**E**) Frequency of the first nucleotide bias of the mature miRNA. Percentage of total base composition of the (**F**) known and (**G**) novel miRNA. Percentage base composition of the first nucleotide in the (**H**) known and (**I**) novel miRNA.

**Figure 2 biology-11-00798-f002:**
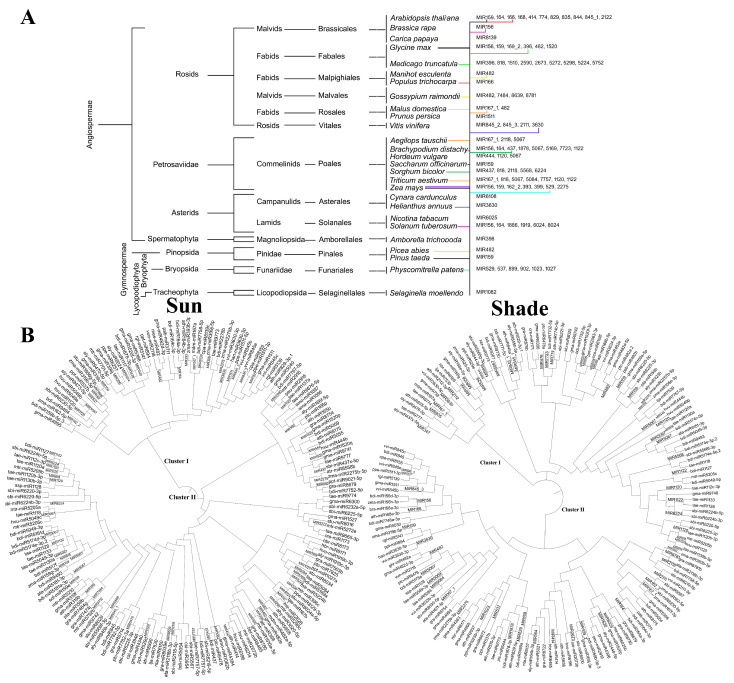
Phylogenetic and cluster analyses of the known miRs under prolonged shade in SP. (**A**) Phylogenetic distribution of 27 genera and 31 species found orthologues in other plants. The phylogenetic tree was built using the common tree tool in NCBI (https://www.ncbi.nlm.gov/Taxonomy/CommonTree/) (accessed on 1 January 2022). Similarity is presented as colored horizontal lines against each MIR. Separate colors are used for different MIR families. miRNA members in each MIR family are presented in Appendix A. (**B**) The cluster-based dendrogram analysis of known miRNA obtained from sun or shade small RNA libraries. Clusters are named in **BOLD** letters, and MIR families are assigned at respective orthologues.

**Figure 3 biology-11-00798-f003:**
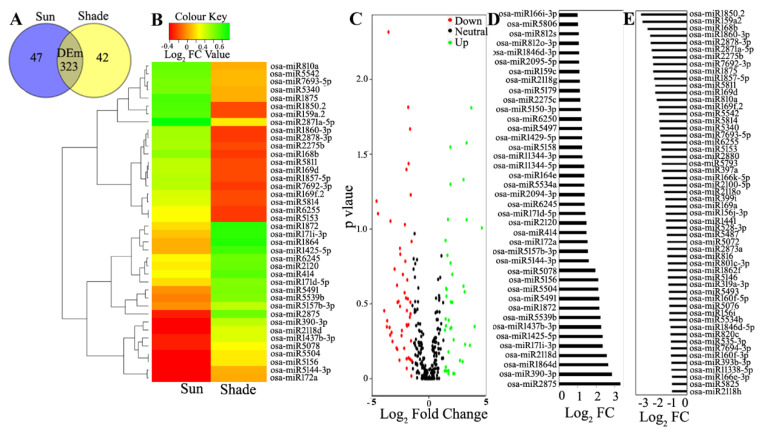
Differential expression analysis of the miRs (DEms) from sun- and shade-grown samples. (**A**) Venn diagram representation of DEms obtained under sun conditions, shade conditions, or commonly found conditions. (**B**) Heat-map analysis of the top 20 up- and downregulated known DEms. (**C**) Volcano plot of the known DEms. Color codes represent the differential regulation of the miRNA. Differential expression of (**D**) 40 up-DEms and (**E**) 54 down-DEms based on Log_2_ fold-change (−1.0 > FC > 1.0 and *p* ≤ 0.9).

**Figure 4 biology-11-00798-f004:**
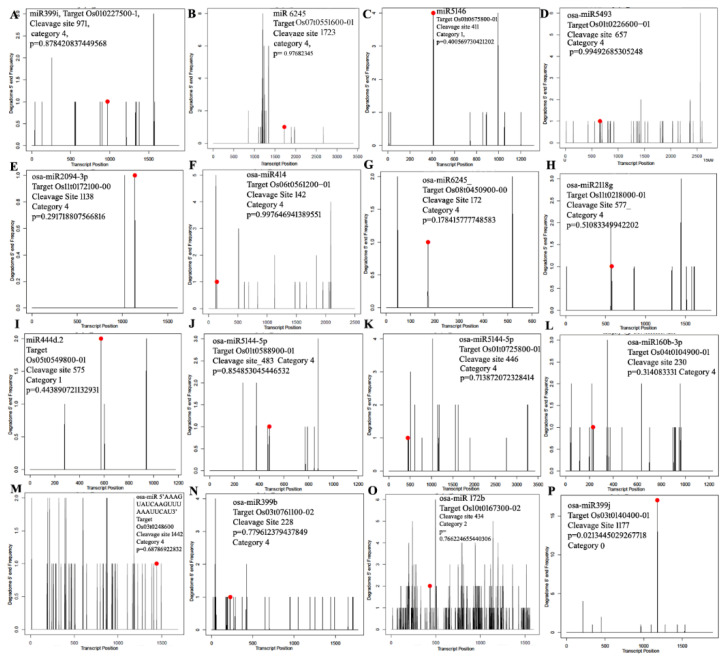
Target plot (t-plot) of 16 targets of miRNA categories including the DEms, neutrally regulated miRNAs, and uniquely expressed miRNAs. Degradome tags including the miRNA, transcript ID, cleavage site, and degradome *p*-value are indicated at top of each t-plot. T-plots were built on the basis of transcript nucleotide vs. degradome 5′ end frequency. The black line represents cleaved products, and the red dot indicates the cleavage site (CS) predicted due to miRNA. The categories (0–4) were based on the relative abundance of the tags at the target sites. T-plot indicating cleavage of (**A**) *Os010227500-1* by *miR399i*, (**B**) *Os07t0551600-01* by *miR6245*, (**C**) *Os01t0675800-01* by *miR5146*, (**D**) *Os01t0226600-01* by *miR5493*, (**E**) *Os11t0172100-00* by *miR2094-3p*, (**F**) *Os06t0561200-01* by *miR414*, (**G**) *Os08t0450900-00*, (**H**) *Os11t0218000-01* by *miR2118g*, (**I**) *Os05t549800-01* by *miR444d.2* (**J**) *Os01t0588900-01* by *miR5144p*, (**K**) *Os01t0725800-01* by *miR5144-5p*, (**L**) *Os04t0104900-01* by *miR160b-3p* (**M**) *Os03t0248600* by miR5′*AAAGUAUCAAGUUUAAAUUCAU*3′, (**N**) *Os03t0761100-02* by *miR399b*, (**O**) *Os10t0167300-02* by *miR172b*, (**P**) *Os03t0140400-01* by *miR399j*.

**Figure 5 biology-11-00798-f005:**
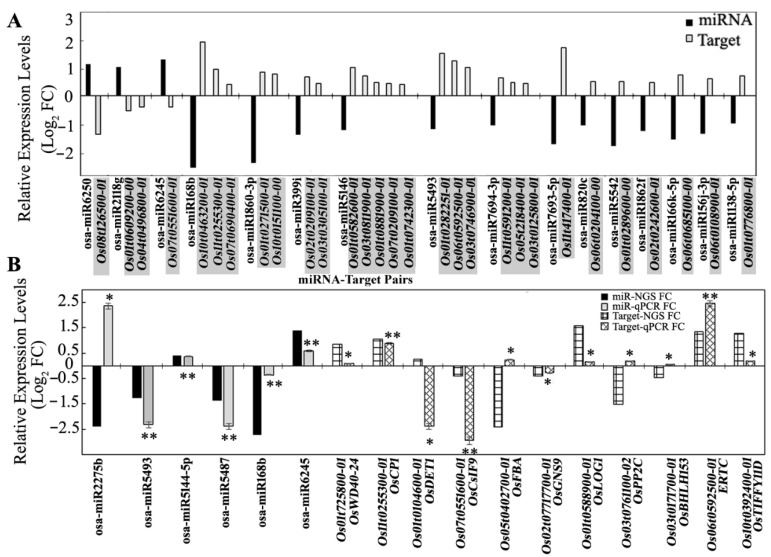
Relative expression levels of DEms and their respective targets. (**A**) Relative expression of top 3 up- and 13 downregulated DEms were obtained from miRNA sequencing. Corresponding cleaved targets were identified with degradome sequencing data. Transcription expression of these targets were filtered from microarray expression data. All data were verified with two replicates for each miRNA or transcript. (**B**) Concordance of relative expression levels of 6 miRs from NGS and 11 targets from microarray were verified using qRT-PCR. Fold change (FC) in gene expression was calculated using the 2^−ΔΔCt^ method. The relative transcript levels are shown as Log_2_ FC. Each data point is a mean of 2 biological replicates performed in triplicate, and the error bar represents the standard error of the mean. * and ** represent significance at *p* ≤ 0.5 and *p* ≤ 0.01, respectively.

**Figure 6 biology-11-00798-f006:**
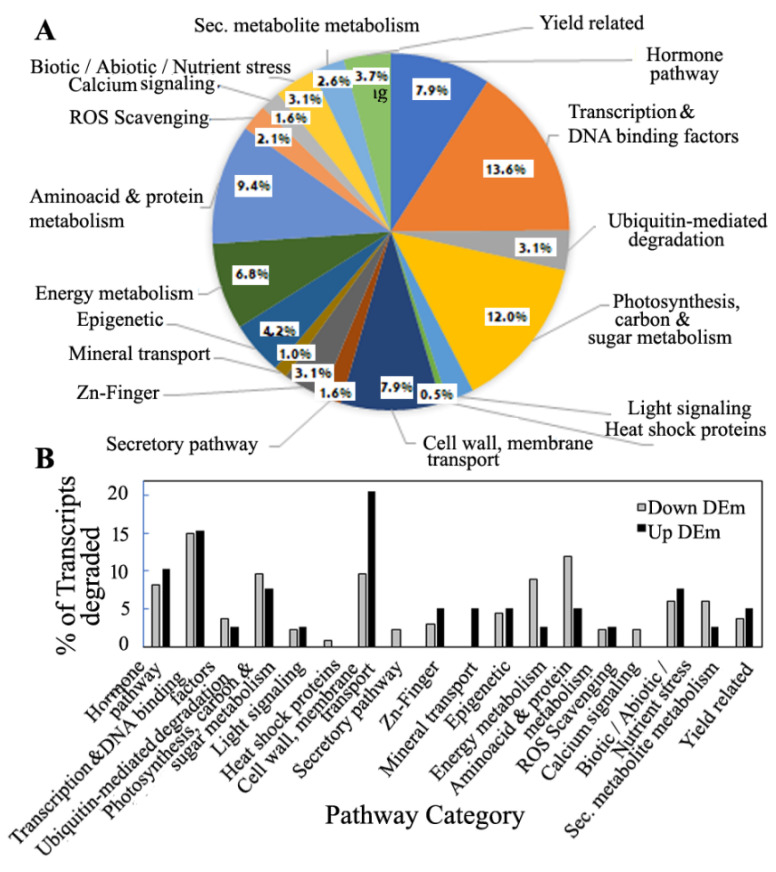
Pathway analysis of the degradome transcripts. Target transcript function was obtained using UniProt database and the RAP-DB and GO annotations. (**A**) Pathway categorization is presented in the form of percentage of the total number transcripts. The total number of transcripts represents transcripts cleaved after removal of hypothetical, unpredicted, and unknown samples, as well as samples with significant log_2_ FC and *p*-values (**B**) Pathway categorization of percentage of transcripts based on up- or down-DEms.

**Figure 7 biology-11-00798-f007:**
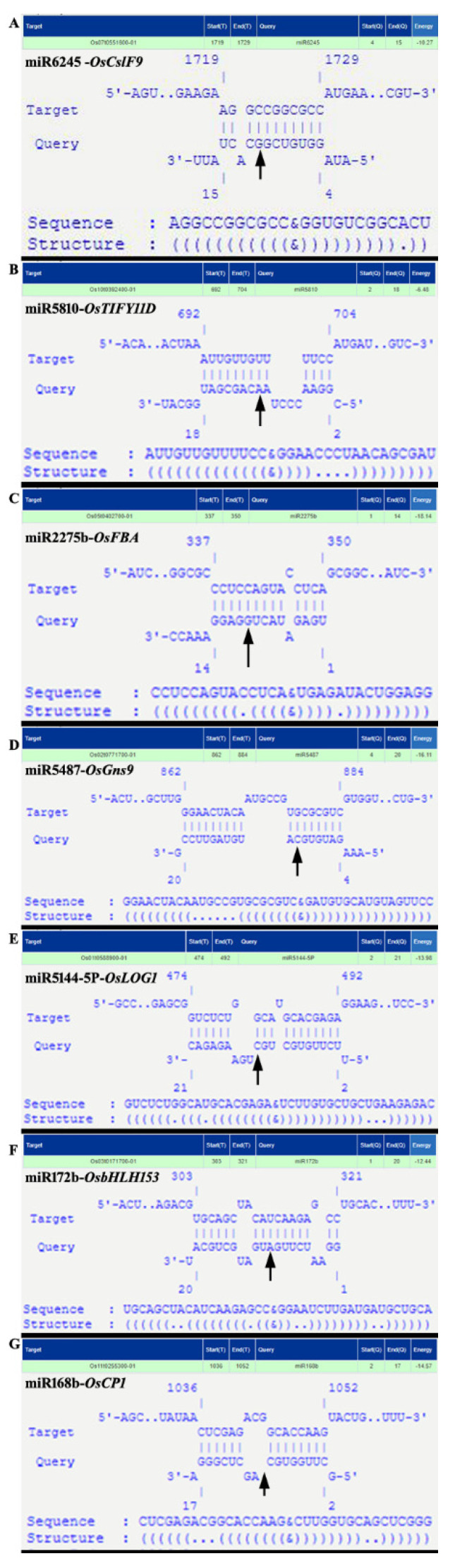
Some of the randomly selected miR–target pairs are presented with their complementary binding and cleavage site. The cleavage sites (CSs) are indicated with black arrows. The complementary binding was observed with the IntaRNA2.0—RNA–RNA Interaction tool (http://rna.informatik.uni-freiburg.de, accessed on 1 January 2022) [60]. The complementary sequence specific binding and CS of (**A**) *miR6245* and *OsCSIF9*, (**B**) *miR5180* and *OsTIFY11D*, (**C**) *miR2275b* and *OsFBA*, (**D**) *miR5487* and *OsGns9*, (**E**) *miR5144-5p* and *OsLOG1*, (**F**) *miR172b* and *OsbHLH153*, (**G**) *miR168b* and *OsCP1*.

**Figure 8 biology-11-00798-f008:**
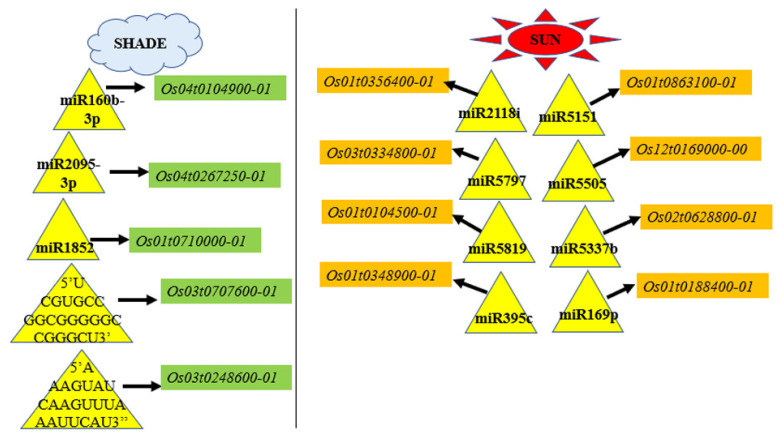
Pictorial representation of transcript regulation by uniquely expressed miRNAs. Represented miRNAs are uniquely expressed under either the sun or shade. However, their expression levels were associated with differential expressions of their targets due to cleavage, as identified by degradome sequencing. Transcript expressions were verified with microarray expression analysis. Yellow color: upregulated transcripts; green color: downregulated transcripts. The black arrow indicate cleavage. Diagram is not to scale.

**Figure 9 biology-11-00798-f009:**
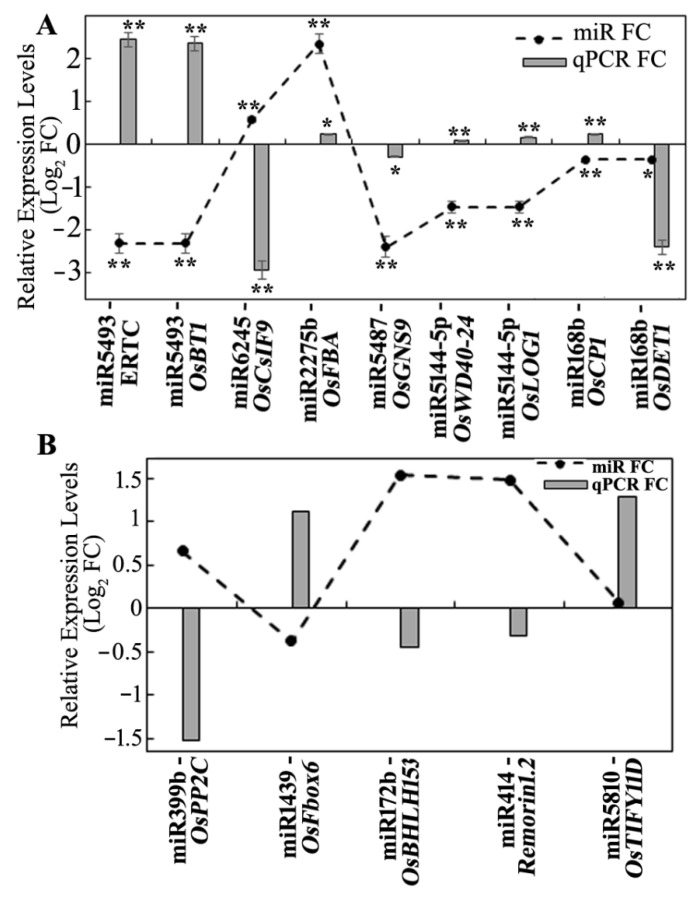
Expression analysis of miRNAs and their targets known to be involved in abiotic stress, light signaling, and shade tolerance. (**A**) Relative expression levels of DEm–target pairs are represented as Log_2_ FC obtained with qRT-PCR. Each data point is a mean of 2 biological replicates performed in triplicate, and the error bar represents the standard error of the mean. * and ** represent significance at *p* ≤ 0.5 and *p* ≤ 0.01, respectively. (**B**) Relative expression levels of DEms were obtained from miRNA sequencing. Corresponding targets were identified by degradome sequencing. Transcription expression of these targets were filtered from microarray expression data. Relative expression levels of DEm–target pairs are represented as Log_2_ FC.

**Table 1 biology-11-00798-t001:** Tabular list of miRNAs and their respective targets identified in this study. These miRNA–target pairs may play significant roles in prolonged shade tolerance in Swarnaprabha. In the miRNA category, the red color indicates upregulated, the green color indicates downregulated, and the grey color indicates neutrally regulated or uniquely expressed miRNAs. Unique expression is indicated within brackets. T-Range: Target range (TStart: alignment start position with transcript; TStop: alignment end position with transcript). Allen score: penalty score calculated per the work of Allen et al. (2005) [58]; MFESite: minimum free energy for alignment in question; MFEPerfect: minimum free energy for perfectly matched site; structure: stable binding of miR and target is accompanied with a decrease in free energy. Aligned secondary structure sequence: aligned sequence.

miRNA	Target ID/Name	Gene Name/Gene ID	Function	T-Range Start-Stop (nt)	Cleavage Site on the Target (nt)	Validated Degradome Category	Allen Score	*p*-Value	MFE Perfect	MFE Site	Sequence	Structure
miR5493	* Os06t0592500-01 *		Heat stress response	239–259	249	4	9	0.999999999998832	−48.2	−32	CGGGGGCGGCGGCGCCCGCGCG&AGC-CGGGCUCUGUCGCGCGUG	((.(.(((((((.(((((.((.&.))-))))).))))))).).))
miR5493	* Os01t0226600-01 *	*OsSLAC*	panicle size and grain yield	646–668	657	4	10	0.99492685305248	−48.2	−34.6	CGCGCGCGGCGGCGGCGGCGGCG&AGCCG-GGCU-CUGUCGCGCGUG	((((((((((((.(((..((((.&.))))-.)))-))))))))))))
miR5493	* Os02t0202400-01 *	*OsBT1-1*	grain formation by controlling starch synthesis	1026–1048	1038	2	9.5	0.957334082969076	−48.2	−33.4	GGCGCGCCGACGUCGGCCCGGCC&AGCCGGGCU-CUGUCG-CGCGUG	.(((((.(((((..((((((((.&.))))))))-.)))))-))))).
miR160b-3p (shade)	* Os04t0104900-01 *	*OsCOMTL2*	methyltransferase	218–240	230	4	7.5	0.69392523364486	−42.8	−29.7	CAUGCUGAGGCUCCUCGCGUCGU&GCG-UGCAAGGAGCC-AAGCAUG	((((((..(((((((.(((.(((&)))-))).)))))))-.))))))
miR169-p (sun)	* Os01t0188400-01 *	*OschlME*	NADP-dependent malic enzyme	1621–1645	1635	4	13.5	0.906795694004373	−39.1	−27.7	GAGCCAGGGUCGUGCAGUAUUUGCC&GGCAAGU-CUGU----CCUUGGCUA	.((((((((....((((.(((((((&)))))))-))))----)))))))).
miR399b	* Os03t0761100-02 *	OsPP2C	stress tolerance. ABA-signalling	217–236	228	4	9	0.779612379437849	−39.6	−26	CGGGGGAGUUCUCGA-UGGCG&UGCCAAAGGAGAAUUGCCCUG	(((((.(((((((..-((((.&.))))...))))))).)))))
miR1439	* Os01t0281000-01 *	*OsFbox6* *OsFBX5* *OsSTA12*	cyclin F-box containing protein	2027–2049	2038	4	9	0.969782287321394	−31.6	−20.7	AUUGCUCAUUCUGUAUUCUGAAA&UUUUGGA—ACAGAGUGAGUAUU	..((((((((((((..(((((((&)))))))--))))))))))))..
miR414	* Os06t0561200-01 *		Potassium/proton antiporter	132–151	142	4	4.5	0.997646941389551	−36.4	−26.8	UCCUCCUCGUCCUCGUCGUU&GACGAUGAUGACGAGGAUGA	((.((((((((.((((((((&)))))))).)))))))).))
miR414	* Os10t0503800-01 *	*OsREM1.2*	Membrane protein, plant growth, development, stress responses	734–753	744	4	3.5	0.809937142647283	−36.4	−30.7	UCGUCGUCGUCGUCGUCGUU&GACGAUGAUGACGAGGAUGA	(((((.((((((((((((((&)))))))))))))).)))))
miR6245	* Os07t0551600-01 *	*OsCslF9*	mechanical Strength of stem	1716–1732	1723	4	11	0.97682345	−19.6	−10.2	AGGCCGGCGCC&GGUGUCGGCACU	(((((((((((&))))))))).))
miR5075	* Os02t0285300-01 *	*OsDREPP2*	plasma membrane protein	361–371	369	4	5	1	−40.2	−27.5	CUCCGCCGCCGUCA-CCA&CGGAUGGCGGCGACGGAG	(((((.((((((((-((.&.)).)))))))).)))))
miR529a	* Os03t0787300-01 *	*OsDjA5*	co-chaperones	1235–1255	1246	4	9	0.999999999991589	−36.7	−24.6	GAGGAGGAGAUGAGGAGGCGG&CUGUACCCUC-UCUCUUCUUC	((((((((((.((((..((((&))))..))))-))))))))))
miR444.2	* Os05t0549800-01 *	*OsEREBP96*	transcription factor in ethylene signalling	564–584	575	1	8	0.443890721132931	−40.1	−26.7	CCGCCGGCGGCGGCGAUUGCA&UGCAGUUGCUGCCUCAAGCUU	..((..(.(((((((((((((&))))))))))))).)..))..
miR5810	* Os10t0392400-01 *	*OsTIFY11D*	jasmonate signalling	692–704	699	4	13	0.999999999324921	−28.98	−6.48	AUUGUUGUUUUCC&GGAACCCUAACAGCGAU	(((((((((((((&))))….)))))))))
miR2275b	* Os05t0402700-01 *	OsFBA	formation of SBP and FBP, increase photosynthetic carbon flux, RUBP regeneration, promote gibberellin mediated root growth	337–350	341	2	5	0.440673027304968	−34	−22.18	CCUCCAGUACCUCA&UGAGAUACUGGAGG	((...(((((((((.((((&)))).)))))))))...))
miR5487	* Os02t0771700-01 *	OsGns9	glycoside hydrolase pollen development, seed germination, cold response	862–884	878	4	20	0.652419432123166	−34.6	−24.6	GGAACUACAAUGCCGUGCGCGUCGUG&GAUGUGCAUGUAGUUCC	(((((((((……((((((((&)))))))))))))))))
miR5144-5p	* Os01t0725800-01 *	*OsWD40-24*	repressors of photomorphogeneis, stomatal closure, mesophyll photosynthesis, sucrose breakdown	435–455	446	4	9	0.713872072328414	−37.9	−25.8	GUCUCGGCAGCAGCGGGUGGA&UUCUUGUGCUGCUGAAGAGAC	(((((..((((((((.(.(((&))).).))))))))..)))))
miR5144-5p	* Os01t0588900-01 *	*OsLOG1*	Cytokinin-activating enzyme	474–492	483	4	6	0.854853045446532	−37.9	−25.8	GUCUCUGGCAUGCACGAGA&UUCUUGUGCUGCUGAAGAGAC	((((((.(((.((((((((&)))))))))))…))))))
miR172b	* Os03t0171700-01 *	*OsbHLH153*	flag leaf angle	303–321	313	4	6.5	0.926716620410872	−36	−24.6	UGCAGCUACAUCAAGAGCC&GGAAUCUUGAUGAUGCUGCA	((((((..((((((((.((&))..))))))))..))))))
miR168b	* Os11t0255300-01 *	*OsCP1*	cysteine protease pollen development	1036–1052	1044	4	6	0.999999999999982	−30	−20.3	ACUCGAGACGGCACCAAG&CUUGGUGCAGCUCGGG	((((((...((((((((&))))))))..))))))
miR168b	* Os01t0104600-01 *	*OsDET1*	repressors of photomorphogeneis	358–375	366	4	6	0.999999999973558	−38.4	−25.9	GCCCGCGCCGCAGCAAGC&GCUUGGUGCAGCUCGGGA	.((((.((.(((.(((((&))))).))).)).)))).

## Data Availability

NCBI Submission details: (i) PRJNA553327 microRNA sequencing of Swarnaprabha in rice (Oryza sativa) bioproject accession no: PRJNA553327, https://www.ncbi.nlm.nih.gov/bioproject/PRJNA553327 (accessed on 1 January 2022); (ii) PRJNA755162 degradome sequencing in Swarnaprabha rice under 75% shade, https://dataview.ncbi.nlm.nih.gov/object/PRJNA755162?reviewer=s005lk4iqtgso329g2tfl3c9nf (accessed on 1 January 2022); (iii) microarray experiment for Swarnaprabha rice line in low light stress (rice). PRJNA485730; GEO: GSE118464. Microarray experiment for Swarnaprabha rice line i... (ID 485730)—BioProject—NCBI (nih.gov).

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
