# Peer review of "Integrated Expression Analysis of Small RNA, Degradome and Microarray Reveals Complex Regulatory Action of miRNA during Prolonged Shade in Swarnaprabha Rice"

_biology, 2022, doi:10.3390/biology11050798_

Round 1
Reviewer 1 Report
The article is recommended to be published.
Author Response
Dear Reviewer,
We thank the reviewer for their kind support and valuable input to improve our MS.

Reviewer 2 Report
I have following comments and suggestion to authors:
- What is the context for using a shade tolerant line for the shade treatment? Wouldn't it be better to compare shade tolerant with a shade senstive line?
- Twenty days of shade treatment seems too long, or is there any reason for keeping it this long?
- Please be carful while using some words for example synteny (line 121). I assume you were using the same line as control and treatment then these lines should be same in terms of genomic sequence.
- While preapring your Figures they should be cited in sequence in the text to avoid any confusion. In your case Figure 1D and others are cited after figure 2A and 2B.
- Figure 2B is not clear, increase the font size and resolution to make it understanbe.
- While reading results I have seen at many places that unnecessary discussion points (Line 197 is example) are included. In the intial of the results just provide some context and then present your results only. Discussion should be in the discussion section.
- In your results, discussion and conclusion; authors has mentioned about the confirmed miRNA-target pair identification. I think miRNA-target pair were identifed not confirmed. Confirmation should be using over-expression and knock-out studies not from expression.
Author Response
What is the context for using a shade tolerant line for the shade treatment? Wouldn't it be better to compare shade tolerant with a shade senstive line?
Response:
Shade-tolerant genotypes have unique variations, alleles, interactions and mechanisms that help them to adapt to prolonged low-light or shade condition. Hence, it was designed to understand the miRNA regulation under shade in comparison to that in the sun in the shade-tolerant genotype Swarnaprabha.
Comparing the mechanisms in a shade-tolerant with a shade-sensitive genotype is also a preferable strategy of validation of shade tolerance. However, the genomic background will vary in that case, which may also affect the type of interaction to adapt to low-light stress. As we think that there can be many types of strategies, interactions and mechanisms to adapt to low-light stress. Hence, to avoid complexity, and to simply understand the miRNA regulations for shade-tolerance we preferred to go with only genotype grown in 2 different light condition.
Twenty days of shade treatment seems too long, or is there any reason for keeping it this long?
Response:
Yes, 20-days of low-light is a long duration stress. However, south-east Asia kharif or rainy season is enriched with overcast cloudy days, which many extend even nearly 10-20 days. Swarnaprabha still maintains sustainable yield in these naturally prolonged cloudy conditions. Hence, we decided to take an extreme and similar conditions to that occurring naturally.
Please be carful while using some words for example synteny (line 121). I assume you were using the same line as control and treatment then these lines should be same in terms of genomic sequence.
Response:
The term is replaced with “collinearity” to avoid any confusion. Line No: 295
While preapring your Figures they should be cited in sequence in the text to avoid any confusion. In your case Figure 1D and others are cited after figure 2A and 2B.
Response:
Presently, Figure 1D is brought before Figure 2A and 2B. For this reason an extra section is made Section 3.1 Line No: 305. Figure 1B and Figure 1C are also rearranged now (Line No:298, Line No: 301) along with changes in the Figure 1. Rearrangement of Figure S1 and Figure S2, Line No: 342, 357 is also done now.
Figure 2B is not clear, increase the font size and resolution to make it understanbe.
Response:
We have increased the clarity and the font size. We have replaced the Figure 2b.
While reading results I have seen at many places that unnecessary discussion points (Line 197 is example) are included. In the intial of the results just provide some context and then present your results only. Discussion should be in the discussion section.
Response:
The concerned lines in section 3.1 Line no 308 and line no 321 in the revised MS are deleted.
In your results, discussion and conclusion; authors has mentioned about the confirmed miRNA-target pair identification. I think miRNA-target pair were identifed not confirmed. Confirmation should be using over-expression and knock-out studies not from expression.
Response:
As suggested, correction throughout the MS is made.
Reviewer 3 Report
The manuscript by Madhusmita Panigrahy et al. provided a comprehensive analysis of differentially expressed miRNAs in a shade-tolerant rice variety, Swarnaprabha (SP), using small RNA seq, degradome, and microarray data. They found the up-and down-regulated differentially expressed miRs’ (DEm) presented drastic differences in the category of targets based on their function and pathway they are involved. They also validated the expression of up-and down-regulated miRNA targets using qRT-PCR. Generally speaking, this is a very well-organized manuscript. The results and conclusion are very clear. Here are just several minor points.
- In figure 1b, there is an apparent high peak at 24 nt of shade replicate 2. While the pattern of other length miRNAs is similar to shade replicate 1. Could the authors add several sentences to discuss the possibility of these results?
- The figure 2b is too blurred to read. I think it may be better if this manuscript is online, but it is not very nice if we read it in printed form. I suggest the authors make this figure larger in the main figures or just put it to supplementary figures.
- For the differential analysis of miRNAs in the sun and shade samples, could the authors indicate the cutoff used to define differential expression in the manuscript?
- In Figure 5, please add the information about the statistical analysis method used in the figure legend.
Author Response
In figure 1b, there is an apparent high peak at 24 nt of shade replicate 2. While the pattern of other length miRNAs is similar to shade replicate 1. Could the authors add several sentences to discuss the possibility of these results?
Response:
Figure 1b represents the length distribution. 2 lines for explanation of higher read count and its significance is added. Line No: 302-304, Page No: 6
The figure 2b is too blurred to read. I think it may be better if this manuscript is online, but it is not very nice if we read it in printed form. I suggest the authors make this figure larger in the main figures or just put it to supplementary figures.
Response:
As per suggestion, we have increased the clarity and the font size. We have replaced the Figure 2b
For the differential analysis of miRNAs in the sun and shade samples, could the authors indicate the cutoff used to define differential expression in the manuscript?
Response:
Log2 Fold-change cut-off is mentioned already in section 2.2, line 217. Now it is again added in results section 3.3, Line 429.
In Figure 5, please add the information about the statistical analysis method used in the figure legend.
Response:
Statistical analysis details are now added in Figure 5 legends. Line No: 581-584
